# Stress, Marginalization, and Disruption: A Qualitative Rapid Situational Assessment of Substance Users and HIV Risk in Lebanon

**DOI:** 10.3390/ijerph19159242

**Published:** 2022-07-28

**Authors:** Kaveh Khoshnood, Amy B. Smoyer, Francesca Maviglia, Janine Kara, Danielle Khouri, Fouad M. Fouad, Robert Heimer

**Affiliations:** 1Center for Interdisciplinary Research on AIDS at Yale, Department of Epidemiology of Microbial Diseases, Yale University School of Public Health, New Haven, CT 06520, USA; francesca.maviglia@yale.edu (F.M.); janinekara@aya.yale.edu (J.K.); robert.heimer@yale.edu (R.H.); 2Department of Social Work, Southern Connecticut State University, New Haven, CT 06520, USA; smoyera1@southernct.edu; 3Independent Researcher, Beirut 1100, Lebanon; danielle.elkhoury@gmail.com; 4Department of Epidemiology and Population Health, Faculty of Health Sciences, American University of Beirut, Beirut 1100, Lebanon; mm157@aub.edu.lb

**Keywords:** substance use, HIV risk, Lebanon, refugees, rapid situational assessment

## Abstract

Lebanon is a diverse and dynamic nation of six million people that has experienced considerable disruption for the last two decades. The Syrian Civil War, which began in 2011, resulted in the displacement of 1.1 million Syrians to Lebanon. Today, Lebanon is the country with the largest per capita number of refugees in the world. In addition, the country experienced a social, economic, and political crisis in 2019 that destabilized the entire society—circumstances that were further complicated by COVID-19 pandemic. With all of the competing calamities in Lebanon, there has been limited scientific investigation into substance use and the risk of HIV infection among the country’s population. To address this gap in knowledge, a qualitative rapid situational assessment (RSA) of substance use and risk of HIV infection in and around Beirut, the nation’s capital, was conducted. The goal of this analysis is to describe the demographics and drug use patterns of this population, explore their HIV knowledge and risks, and build knowledge about their perceptions of and access to substance use treatment and other social services.

## 1. Introduction

This review of the literature summarizes what is known about substance use and HIV in Lebanon, including a focus on research about the refugee community within the country.

### 1.1. Substance Use and HIV Risk in Lebanon

Available data on the prevalence of substance use among the Lebanese citizens is limited and much of it is dated. In 2003, the prevalence of lifetime drug use among persons aged 18–44 years old was 0.6% and the 12-months prevalence in the same group to be 0.3% [1]. Research by the World Health Organization about Lebanese high school students reported that, in 2017, 2.1% of students aged 13–17 had ever used marijuana [2]. Finally, data on the number of persons in detention centers due to drug-related charges show a 108% increase between 2011 and 2018 [3].

HIV prevalence among adults aged 15–49 in Lebanon is below 0.1% [4]. The main mode of transmission is through sexual relations, and research has reported prevalence rates among men who have sex with men (MSM) between 1.5 and 12% [5,6]. At 0.9%, the prevalence of HIV among people who inject drugs (PWID) is higher than that of the general population but relatively low compared to rates among PWID in most other countries [6,7].

Research about substance use and HIV risk in Lebanon is also limited, and studies have found conflicting results. Some evidence suggests a high prevalence of HIV risk behaviors, such as sharing of syringes, among PWID [8]. In 2007, research about PWID (*n* = 212) in the Beirut area found that 65% had some knowledge about HIV transmission and prevention; however, only half recognized that sharing syringes could lead to infection and only 30% reported using condoms consistently over the past month [9]. More recent research by UNAIDS estimated that 98.5% of PWID in Lebanon followed safe injection practices [7], and a systematic review of HIV knowledge in the Middle East and North Africa found that PWID in Lebanon had a near universal knowledge of dynamics of HIV transmission [10].

### 1.2. Substance Use and HIV Risk among Displaced People

Substance use and HIV risk among displaced populations is a neglected area of public health. A 2016 global systematic review on alcohol and substance use among forced migrants found that only 6 out of 63 studies estimated a prevalence of substance use disorders [11]. Comparative analysis has found that refugees living in community settings have lower rates of substance use when compared to people living in camp settings [11]. This seems to be the case in Lebanon, where a recent cross-sectional study that examined substance use among Palestinian and Syrian residents in three camps in Lebanon found a higher rate of lifetime substance use among Palestinians born in Lebanon compared to those displaced from Syria [12].

Research has demonstrated that pre- and post-migration stress and trauma are risk factors for substance use among refugees and populations in humanitarian crisis settings [13]. The lifetime prevalence of PTSD was found to be 35.4% among Syrian refugees living in informal tent settlements (ITSs) in Lebanon [14]. In other contexts, PTSD diagnosis is associated with increased rates of substance use disorder among refugees [15].

To our knowledge, no study has been conducted to assess prevalence and knowledge of HIV among refugees in Lebanon. In other Middle Eastern countries, research has found that refugees and migrant workers have less HIV knowledge compared to resident populations [10]. While the prevalence of blood-borne infections among refugees and migrants is, overall, an under-researched topic, a concentrated epidemic of HIV among Afghan refugees in Pakistan was reported in 2010 [16].

### 1.3. Gaps in Knowledge and Study Goals

This paper reports on findings from a Rapid Situation Assessment (RSA) of substance users in Lebanon, including Lebanese nationals/permanent residents and refugees. The goals of this study are to describe the demographics and drug use patterns of this population and explore their HIV knowledge and risks. This study also addresses existing gaps in the literature by building an understanding about substance users’ perceptions of and access to treatment and other social services. As such, it identifies the needs of drug using populations that can inform the development of targeted interventions. Finally, COVID-19 and Lebanon’s political and financial crisis of the past two years have caused widespread economic precarity and food insecurity among Lebanese and displaced populations alike [17]. This study begins to document some of the effects of these national crises on people using drugs in Lebanon.

## 2. Materials and Methods

Data for this analysis come from individual interviews conducted in and around Beirut (Lebanon) with adults with a history of substance use. The Consolidated Criteria for Reporting Qualitative Studies (COREQ) checklist, “developed to promote explicit and comprehensive reporting of qualitative studies”, is used to describe the project’s methodologies [18] (p. 356).

### 2.1. Research Team

All of the interviews were conducted by staff working at the two community-based health care agencies with a focus on people who use drugs. The data collection team included a psychiatrist (male), a social worker (woman), and a psychologist (woman). The interviewers, who identified as Lebanese nationals, spoke both English and Arabic. Team members underwent a two-day training and were trained in person by the first, fourth, fifth, and the sixth author in preparation for data collection. This training included information about human subject research ethics, the interview methodology, and the project’s research questions and goals. In many cases, the interviewer–participant pair had an existing staff–client relationship.

### 2.2. Study Design

Given the shifting socioeconomic and political vulnerabilities in Lebanon, the study was designed as a Rapid Situational Assessment (RSA) that uses qualitative and quantitative methods to construct a quick assessment of a situation to inform healthcare programs and interventions [8,19]. An RSA can be used to answer a wide variety of research questions, from assessment to evaluations, generating information “from multiple sources as quickly as possible” [19] (p. 7). This RSA sought to understand participants’ psychosocial circumstances and the impact of these circumstances on their access to healthcare, including substance use prevention and treatment. Participants’ knowledge and attitudes about HIV were also assessed. Thematic analysis was used to manage, organize, code, and describe the qualitative data that was collected through these semi-structured individual interviews [20].

Conducting research during a humanitarian crisis is complicated, which contributes to the paucity of research about drug use and HIV risk in these circumstances. In designing this study, every decision prioritized the safety of the participants and staff, even when these decisions introduced methodological challenges that would complicate data collection and analysis. Any inconsistencies or misalignments between the research methods and the stated purpose and goals of this project should be understood not as careless mistakes but as deliberate choices made in light of the necessary realities of the circumstances in which the inquiry unfolded.

The methodological challenge posed by this RSA concerned recruitment. Early on, the team decided to avoid further marginalization of stigmatized communities by including Lebanese nationals, Lebanese residents, and foreign-born displaced people, including refugees. Further, direct community-based recruitment of participants was not feasible given the criminal-legal issues associated with engaging with people who use drugs and displaced stateless people in Lebanon. From this, it was necessary to limit the sample frame to people who could be engaged by one of the two non-governmental organizations (NGOs) that collaborated on data collection. The criteria that the participants must report “a history of drug use” was also intentionally vague to protect participants and resulted in a sample that included middle-aged people with decadelong histories of heroin addiction and young adults who occasionally drank beer or smoked hash. The result is a patchwork of stories with diverse sizes, shapes, and textures that are sewn together into a single quilt.

#### 2.2.1. Participant Selection

Convenience sampling was used to recruit participants at two community-based agencies in Beirut. Agency A provides substance use treatment, including opioid substitution therapy (OST) and other harm reduction services, HIV/AIDS prevention services, counseling and rehabilitation, and other health-related services. Most of the clients at Agency A are Lebanese nationals or residents. Agency B provides psychosocial and health services to people living in residential areas (“camps”) designated for refugees outside of Beirut. Most of the clients at Agency B are Syrian or Palestinian. Clients at Agency A are, by definition, people who use drugs, while people who engage with Agency B may or may not identify as drug users.

Forty (40) participants were recruited through each agency. Eligibility criteria required that participants be 18 years of age or older and have “a history of drug use”. At Agency A, participants were clients at the agency who were approached by staff and asked if they would be willing to participate in a study about substance use, HIV, and health. At Agency B, flyers, distributed at the agency and in the refugee camp in Arsal surrounding the agency, invited people to be interviewed about substance use, HIV, and health. Arsal is a small Lebanese town in Bekaa valley on the border with Syria, known for harsh weather, especially during winter, and for hosting thousands of Syrian refugees, who live in informal tented settlements and inadequate shelters. Lebanese citizens in Arsal are also very poor. Both communities are served by a few health centers funded by charities and NGOs, in addition to a mobile clinic. The Arsal area is under Lebanese army control because of security issues related to instability in Syria. Participants in the Agency B sample included clients and people who were not clients of the agency but who lived in the surrounding geographical area. All participants received compensation of USD 20 for their time.

Clients who expressed interest in participating were screened by the agency staff to determine if they met the study’s inclusion criteria. If they qualified for the study, a time, date, and location for data collection was scheduled. The same people who conducted the screenings also conducted the interviews. Interview locations were identified based on the participants’ preference and convenience: 42 interviews took place at community-based agencies, 18 interviews took place in a camp designated for refugees, 13 interviews were conducted at a hospital, 4 were conducted in participants homes, and 3 were conducted by telephone. Locations were agreed upon by both parties and interviews were carried out with confidentiality. All the individuals who volunteered for the interviews completed the data collection (there were no refusals or dropouts). The names used in presenting the data are all pseudonyms assigned by the authors.

#### 2.2.2. Data Collection

At the time of these appointments, verbal informed consent was administered and then the interview was conducted. As described earlier, the interviewers had been trained by the first author on human subject research ethics and the informed consent process. Interviews were conducted in Lebanese dialect/Arabic based on the participants’ preference and digitally recorded. During the interview, the interviewers recorded handwritten notes. After the interviews were completed, the interviewer reviewed the audio tape to expand and elaborate upon their notes and conducted some “spot transcription” for responses that they found to be particularly descriptive. During this review process, all notes were recorded in English. The interviewers translated the interviews into English during the note review process. In keeping with RSA format and goals, verbatim transcription and translation were not conducted. The data presented here approximate the clients’ narratives in order to share the speakers’ thoughts and knowledge but may not reflect their specific words.

The semi-structured interview instrument included 30 questions. About half of these questions were close-ended questions that could be answered in one sentence or word (e.g., where were you born?) and half were open-ended questions that invited more detailed responses. On average, interviews took 38 min to complete, with times ranging from 15 min to 75 min. The interview began with socio-demographic questions, including questions about national identity and history of displacement, if any. Next, a series of questions about health and drug use history were explored to understand participants’ perceptions of their health and experiences accessing care. Alcohol and drug use experiences, habits, and behaviors were discussed. Participants were also asked about their knowledge, perceptions, and testing history related to HIV. Finally, participants were asked to reflect upon how the current socio-political crisis had impacted their health.

Thematic analysis was used to identify key themes in the data. A set of a priori codes was developed by the first and second authors based on the interview instrument. The second author then coded 15 of the interviews using these a priori codes and added in vivo codes that surfaced during this analysis. This initial review included data collected from both Lebanese and refugee respondents. The revised codebook that reflected this open coding process was then reviewed by the first and second authors, and codes were consolidated and defined. Next, 10 additional interviews were coded with this updated codebook and then this work was again reviewed by the first and second authors. The final codebook included 13 codes and 65 sub-codes related to psychosocial history, substance use, health access, and HIV. These codes were primarily objective in that they categorized and organized the data, rather than analyzing or interpreting the narratives. The third author coded all 80 interviews using the full codebook.

In preparation for this manuscript, the second author reviewed the coded data, and added four additional normative codes for this specific analysis: interpersonal stress, structural stress, marginalized, and belonging. The first and second author then coded all 80 interviews using these new codes. These findings were reviewed with the third author and then discussed with the fourth, fifth, and sixth authors, who are Lebanese and Syrian scholars. Discussion with these key stakeholders, who were also involved in designing the study, served as a form of member-checking. Their input suggested that the study findings aligned with their experiences with and knowledge about people who use drugs in Lebanon.

## 3. Results

Participants’ narratives about their lives, the drivers of their substance use, access to and experiences with health care and substance use treatment, and knowledge about HIV paint a picture of diverse and engaged communities trying to survive in a complicated low-resource environment. Interpersonal and structural stress is a dominant theme that surfaced in all the participants’ narratives about their current lives and substance use behaviors. Among both Lebanese nationals/residents and refugees, there was a sense of being untethered. Narratives about marginalization and belonging were also articulated.

### 3.1. Description of Sample

#### 3.1.1. Socio-Demographics

The 80 participants included Lebanese nationals (*n* = 49), refugees (*n* = 24), and non-citizen permanent residents (*n* = 7). Four of the seven had Syrian nationality and three identified as Palestinians. The refugees were all Syrians displaced by the Syrian Civil War, with the exception of one Iraqi woman. Ten of the 24 refugees were registered with the UNHCR, four were unregistered, and ten did not clearly define their refugee status during the interview. Respondents included 71 cisgender men, five cisgender women, three transgender women, and one transgender man. Two participants were under 20 years of age. The average age of participants was 31.8. Most were between the ages of 20 and 29 (*n* = 35) or 30 and 39 (*n* = 31), with a handful in their 40s (*n* = 8) or over 50 (*n* = 4). There was no difference in the average age of the Lebanese nationals/residents and the refugee participants. However, their living situations were quite different. Most of the Lebanese participants were living with their parents and siblings while the majority of the refugee participants lived with their wives and children.

#### 3.1.2. Substance Use and Treatment

The sample included a spectrum of drug use behavior from opioid injection to alcohol consumption. Nearly half of the participants (*n* = 38) reported either current injection drug use (*n* = 13) or ever having injected drugs (*n* = 25). Almost all (*n* = 36, 95%) of these people with a history of injection were Lebanese citizens or permanent residents. In addition to injection drug use, participants reported consuming opioids, “party drugs” (e.g., ecstasy/MDMA), psychedelics (e.g., LSD or mushrooms), cocaine, painkillers (e.g., tramadol and ketamine), benzodiazepines (e.g., lexotanil, rivotril, and valium), cannabis, and alcohol. In contrast, only 2 of the 24 refugees who participated in this study reported ever injecting drugs; most of these individuals primarily used alcohol and/or cannabis, and many did not identify their drug use as problematic. The differences in patterns of substance use that surfaced in this qualitative sample cannot be generalized; they reflect the study’s recruitment strategies and differences in the mission and client-base of the agencies that collaborated on data collection. The Lebanese clients at Agency A were, by virtue of their engagement with this agency, opioid users, while the refugees interviewed by Agency B were either not receiving services from the agency or were clients receiving general health services.

For the same reasons, substance use treatment history was common among the Lebanese nationals and residents who were interviewed by Agency A, while only one of the refugees had any experience with or access to treatment. This lack of treatment history aligns with these participants’ understandings of drug use as social and non-problematic. Ibrahim (Syrian, male, 25 years old) remarked, “I never asked for help. There was no need. I never perceived myself as an addict. But smoking is the only way I cope with this difficult life”. Alcohol and cannabis use was normalized by refugee participants, and they were not seeking support to curb their use.

#### 3.1.3. HIV Knowledge and Testing

Most participants recruited had a broad and general understanding of HIV, identifying sexual behavior, needle sharing, and mother-to-child transmission risks. However, there were several Syrian men in their 20s who had never heard of HIV. Further, there was some confusion about the risks associated with sexual behavior and whether the virus could be transmitted through saliva, toothbrushes, or touching. Less than half of the participants knew there are effective treatments for HIV (*n* = 37). About half of the participants had ever been tested for HIV (58%, *n* = 46). Most of the people who knew about HIV risk and had been tested, and the four people who self-identified as living with HIV, were Lebanese nationals/residents. Testing was low among refugees (17%, *n* = 4), and none of them identified as HIV-positive. Again, the differences in their substance use patterns and engagement in care, due to the sampling strategy, may explain these differences.

All of the participants who reported currently injecting drugs stated that they do not share syringes with others, although some reported sharing in the past and currently having to reuse their own syringes due to scarcity. Participants reported accessing syringes through pharmacies and syringe exchange programs. Given that these individuals were clients of harm reduction agencies, their access to sterile syringes is not surprising and cannot be generalized to all people who inject drugs in Lebanon.

### 3.2. Thematic Analysis

In discussing their lives and experiences of drug use and treatment, narratives circled around a common theme of stress and coping. Participants relied on substance use to cope with both interpersonal and structural stressors. The other theme that surfaced in this data was participants’ experiences of marginalization and desire to belong.

#### 3.2.1. Interpersonal Stress

Participants reported using substances to cope with difficulties in interpersonal relationships (i.e., romantic relationships, friendships, school or work relationships, family dynamics). For example, Amir (Lebanese, male, 27 years old) started binge drinking following the death of a close relative and difficulties with intimate relationships. Daiam (Lebanese, male, 23 years old) also reported using substances to self-medicate: “I have a pain inside of me repressed because of my childhood experiences mainly and I think that psychedelics help us get free from our pain”. Participants described using drugs to help manage social anxiety at parties and nightclubs and cope with conflict in their homes. Mubariz (Lebanese, male, 26 years old) said he started using because “My parents used to fight a lot! Over anything in life! My father used to cheat on my mother and she used to cry a lot!”.

The decision to self-medicate reflected a lack of confidence in mental health interventions: “nothing but drugs will ever help you deal with your issues”. (Hani, Lebanese, male, 26 years old). Dasia (Lebanese, female, 23 years old) also suggested that substances were the best solution for psychological and emotional issues: “There are two ways to deal with problems in life. People tend to either drink alcohol and use drugs or commit suicide! You have no other choice!”. Iman (Lebanese, male, 25 years old) shared a similar position: “I encourage people to drink and use alcohol to avoid dying due to stress”.

Workplace stress was also a trigger for drug use. Substance use helped mask interpersonal stress and physical pain related to employment. Maheer (Lebanese, male, 32 years old) used alcohol and hash to develop the patience required to handle customers at his cell phone store and Cyla (Lebanese, transwoman, 34 years old) relied on a variety of drugs in order to engage in sex work. Similarly, several Syrian respondents reported consuming drugs to manage the physical pain caused by heavy manual labor. Mahmoud (Syrian, male, 26 years old) reported using cannabis and captagon “to get me through my days, I can’t work without them, I have to take them daily because my work is very hard”.

Financial problems aggravated participants’ interpersonal stress and substance use. I started [to use heroin] at an advanced age, at the age of 38 years old. It was a very difficult period in my life. I discovered that my son is autistic, and I couldn’t find an institution for him because of my limited financial means. One friend told me that he can bring me something that can make me feel good. He brought me heroin. (Kashir, Lebanese, male, 43 years old).

While Moustafa describes an acute situation that triggered his substance use, Masab (Lebanese, male, 27 years old) spoke about a constellation of persistent circumstances that led to his heroin use: “I am not working and I don’t have anyone intimate in my life and there are always problems surrounding me and brother in jail and can’t do anything for him”. In this narrative, he links his financial insecurity, resulting from his unemployment, to his inability to find a partner and support his incarcerated brother.

For refugees, substances helped them to cope with emotional pain and the trauma of displacement. Many refugees spoke about using drugs to escape. Emir (Syrian, male, 25 years old) talked about escaping his memories:


*I am far from my family and my country. I left everything and I lost everything … my life, my house, my family … and I won’t be able to go back. I started smoking hashish here four years ago. When I was in Syria, I never thought about drinking alcohol or smoking hashish, but now I feel sometimes that I need to, especially when I feel nostalgic.*


Mohammad (Syrian, male 28 years old) framed his escape a bit differently. For him, substance use allowed him to escape his current reality, living with his wife and children in a refugee camp.

I started smoking hashish when I came to Lebanon. I discovered it here and it was an escape for me. I smoke one joint every 3, 4 days. I drink arak or whisky once per week and I put cologne on my cigarettes, 2 to 3 packs a day. It doesn’t help me feel high, but it helps me escape the reality a little bit, for a very short period.

As both of these men describe, many refugees began to use drugs after their forced displacement. Isolated in a new setting with reoccurring thoughts about trauma and loss, refugees reported using drugs, especially cannabis, to escape these painful emotions.

#### 3.2.2. Structural Stress

There were two major structural determinants of stress described by the participants. For the Syrians, the socio-political circumstances that provoked the Syrian Civil War and forced them to flee their country created a vulnerable reality that increased substance use. For all of the participants, and especially the Lebanese people, the socio-economic crisis of 2019, and the COVID-19 epidemic that followed, were structural stressors that increased substance use and interrupted treatment.

The Syrian men talked in detail about the traumatic events of the war and their displacement, sharing harrowing stories about escaping violence on foot and by motorcycle, and the continued trauma inflicted by the harsh conditions in the camps. Khaled (Syrian, male, 40 years old) reports consuming alcohol and cannabis with other Syrian men in the camp to cope with the structure violence they have endured:


*Running and escaping for our lives, how do you imagine that was like? Utterly humiliating. At first, I came through a path on a mountain bike and then assured a car and a safe passage for my wife and kids in a car. It was very stressful and confusing. Oh well better than dying! There’s no work but getting by and there are some good people and NGOs that help get us by.*


Habib (Syrian, male, 24 years old) also reports smoking cannabis to cope with life in the camps:


*Living in Lebanon isn’t easy because of the poverty and because the lack of work opportunities … Rainwater flows inside the camps when it rains and in summer it is super hot. The hygiene is zero. We don’t have proper toilettes. There is no access to health services.*


Because most Syrian participants attributed their drug use to the deprivation of refugee life, many believed that returning to their country would be the only way to halt their consumption: “If the regime changes and I go back to my country and have a chance of a decent life and a good future I would definitely stop” (Mahmoud, Syrian, male, 26 years old). As long as they continued to live in the camps, however, participants reported that they had little motivation to stop using: “What can improve and help us decrease drug use is going back to our country and feeling better psychologically. Nothing else can be done” (Ibrahim, Syrian, male, 25 years old).

The harsh socio-economic downturn in Lebanon that began in summer 2019, and the disappointment and frustration produced by this situation, was another a source of stress that participants reported as having increased their drug use and decreased access to treatment. Hassim (Lebanese, male, 61 years old) said “I feel desperate and hard to find small jobs in electricity or small tasks, so less money for food and I economize my substitution treatment pills, my bup[renorphine], so it lasts more, [but] I feel more frustrated and hopeless”. The boredom and “unproductivity” of unemployment also sparked drug use, as people sought to “fill their time” (Farez, Lebanese, male, 30 years old; Mohamed, Palestinian, male, 34 years old). Participants’ efforts to resist substance use were complicated by interruptions in services, including healthcare and public transportation, due to the national economic-political crises and COVID-19. Jean-Marc remarked “Everything is closed in the country except the dealers! They’re making good money now!” (Lebanese, male, 24 years old).

These crises were not, however, universally detrimental to people’s substance use patterns. Akkuna (Lebanese, male, 49 years old) was emphatic that nothing had changed, stating “I’m still accessing services in the same way and I’m still using alcohol and drugs in the same way”. Similarly, some Syrian participants reported, that because the circumstances in the camps were already horrible, their lives were unchanged by the crisis. Others reported that their lack of income and constrained mobility decreased use. Naser (Syrian, male, 27 years old) explained that “because of the situation I cannot afford to go out and do what I used to do before the revolution”. Moustafa (Lebanese, male, 43 years old) described spending more time with his children and walking for hours in order to manage his withdraw symptoms.

#### 3.2.3. Marginalization

Marginalization was a common theme across the participants’ narratives. As Abbas (Lebanese-Syrian, male, 23) described, “People here deal and communicate with each other depending on their religion, sexual orientation and nationalities”. As an HIV positive gay man with a Lebanese mother and a Syrian father, Abba experienced intersecting oppressions that limited who would be willing to “deal and communicate” with him. Participants also described the ways in which their status as substance users marginalized them from society at large and from systems of care in particular. Some participants felt deliberately excluded from treatment and health services, reporting that their applications for care were rejected without explanation or lost in bureaucratic systems. For those who managed to enroll, rigid agency policies, especially about attendance, interrupted care: “I used to be late to meetings and psychologists, so they kicked me out. It wasn’t a nice way to kick out a patient, and I had to stop the OST and they didn’t give me transfer papers. They are the reason I relapsed” (Lena, Lebanese, female, 28 years old). In this story, Lena describes being “kicked out” not only from her program, but from the entire system of care.

Refugees described being socially marginalized by Lebanese people who “make us feel like intruders”, treated themselves in a “superior way”, and “like lesser creatures” (Mahmoud, male 26 years old; Emir, Syrian, male, 25 years old; Ghazi, male, 32 years old). In his narrative, Ahmed (Syrian, male, 18 years old) offered a more nuanced perspective of Lebanese reaction to Syrians:


*Some people are good and treat us in a fair way, but other treat us in a superior way and make fun of us because we are Syrian. It is hurtful because we are all humans and equal and what they don’t understand is that we are here by force, we all wish to go back to Syria, but we can’t because we are against the regime.*


The legal status of the participants shaped perceptions of their place in Lebanon. Those who did not have an official refugee status felt particularly vulnerable. Dani (Syrian, male, 28 years old) explained it in this way:


*Thanks God I am here with my brother. We help each other a lot …. And what worries me is that I still don’t have the UN card, but my brother has it and like I told you we help each other a lot … Sometimes I feel so frustrated and helpless because of the situation.*


Other displaced families experienced a more complete marginalization because all members were without refugee status. Falah (Syrian, male, 18 years old) came to Lebanon with his parents and siblings: “We are not registered in the UNHCR program so we get no help”.

In addition to being partially or fully excluded from Lebanese society and social services, refugees also experienced marginalization in a very literal way: The geographic location of their homes in refugee camps complicated their ability to access drug treatment programs: “No programs here [in camps] for drugs or anyone to talk to, you don’t really get helped if people know about you” (Saleh, Syrian, male, 48 years old). By describing how people cannot even find “anyone to talk to”, Saleh narrates his socially constructed positionality as an “other”. The larger society’s inability to even hear refugees’ concerns leaves displaced people feeling forgotten and overlooked.

#### 3.2.4. Belonging

In their narratives, participants described a desire to belong to community and family, and situations where they felt this sense of fit. Participants linked this need to belong to substance use: “When the community is better and you can work like a decent person and be regarded in a good way and have a good life and feel safe, you don’t smoke any more, you don’t need it” (Samaan, Syrian, male, 26 years old). Participants endorsed the idea that recovery requires changes in mindset and lifestyle, cutting ties with negative individuals, keeping busy, and connecting to community through employment, volunteering or sports. From this, efforts to reduce stigma and allow space for people who use drugs to join society were described as critical. While most participants described treatment providers as inclusive and non-judgmental, Eessa (Lebanese, male, 29 years old) noted that stigma could exist even within treatment programs: “Once I had a positive urine test, so they threatened me. I didn’t like that because I am here to get treated and not punished”. Participants greatly appreciated the interpersonal support they received from family, friends, and service providers: “When one stands alone, he doesn’t have anyone to lean on, or share their difficulties with. [Treatment is more successful] if one had his family members accepting who he is, respecting his life choices and trying to help instead of judging him” (Yousef, Lebanese, male, 28 years old). Several participants call for the reform of criminal-legal policies that institutionalize exclusion, especially policies related to background checks. Ali (Syrian, male, 39 years old) asserted that treatment services were useless without “shelters, health support, and respect. To what extent would it [treatment alone] help people? I’m sorry I don’t mean to be harsh, but this is the reality!” In calling for both specific services and “respect”, Ali expresses a desire not simply to be served, but to be included.

This highlights the need for belonging, which was particularly pronounced among the Syrian refugees. Some people found a sense of belonging in the camps: “Here in [the camp] it’s like we are one big family. I feel good and safe. People don’t treat me in a strange way. But outside this village, yes, so I don’t leave it” (Khaled, Syrian, male, 40 years old). Employment was another way that people connected with community: “With the décor work I do and for cheap all the Lebanese in the area love me and my work” (Irfan, Syrian, male 48 years old). Participants articulated a need for comprehensive services that recognized their humanity and allowed them to fully engage with the larger society.

## 4. Discussion

These findings, generated from a convenience sample of Lebanese and refugee people in Beirut (primarily young adult and middle-aged men) with histories of substance use, describe patterns of drug use among this population, explore their HIV knowledge, and build understanding about their perceptions of and access to social service healthcare systems. The findings also inform discussion about what resources are needed to address substance use and HIV risk in Lebanon and other parts of the world impacted by socio-economic and political disruption. Further, this data suggest the critical importance of the way these services are delivered. Participants’ responses indicate that a wide range of substances are available and consumed in Beirut and the refugee camps outside of the city. While access to substances was more constrained in the camps, alcohol, cannabis, pills, and heroin were reported. In their narratives, participants describe the interpersonal and structural stressors in their lives and the ways in which this stress impacted their substance use and treatment access. These findings about patterns of substance use and stress, especially related to the political and social unrest, economic deprivation, and the trauma of displacement, are consistent with research from other contexts [11,15,21]. Taken together, these narratives of stress and disruption paint a complicated picture of HIV risk in Beirut.

### 4.1. Stress, Substance Use, and HIV Risk

The overwhelming stress of participants’ daily lives was the main story that participants sought to recount. Participants used substances to cope with interpersonal and structural stressors in their lives. For the most part, the HIV risk associated with this substance use (as described by participants) was minimal. Many people, especially in the refugee community, reported never having injected drugs. People who injected drugs expressed a commitment to using sterile syringes and had access to this equipment through pharmacies and harm reduction agencies. However, this HIV safety seemed tenuous, as descriptions of access to sterile syringes, counseling, and OST had been interrupted when the socio-economic crisis jeopardized public transportation systems and their ability to make co-payments. Moreover, the possibility that the substance use patterns of refugees could intensify in the face of prolonged displacement is conceivable. If injection practices were to increase in these camps, their lack of knowledge about HIV risk and lack of access to treatment and harm reduction services could produce an environment of high HIV risk.

### 4.2. Sexual Behavior and HIV Risk

While knowledge of HIV transmission was high, misconceptions and inaccuracies persisted, especially about sexual behavior and casual contact. Most of the Lebanese nationals and residents were unmarried, and their narratives suggested that they were sexually active, trying to build intimate relationships, and/or interested in marriage. Their sexual risk assessment and prevention strategies were not clear. Several participants identified as sex workers. More in-depth discussion about sexual behavior (including sex work) among heterosexual and gay adults in Lebanon, and the impact of the economic crisis on individuals’ ability to negotiate condom use, is needed to fully understand this community’s HIV risk. The majority of the refugees were married and had multiple children, indicating that they had engaged in unprotected sex. Refugee participants’ beliefs that their marital status protected them from HIV reflected a confidence in marriage as a monogamous relationship that may not be merited. Especially in light of the drug use, severe poverty, and economic hopelessness shared in these narratives, the possibility of transactional and non-transactional extra-marital sexual relationships seems possible. The lack of concern about HIV among refugees in this sample underscores the urgent need for research about sexual behavior in the camps to develop sexual health education and services among displaced populations.

### 4.3. Belonging and HIV Risk

Most participants described feeling alienated or marginalized by society. Lebanese nationals and residents who were substance users felt stigmatized and rejected by community, including family and healthcare providers. Refugees described living at the margins of community in both a physical, geographic sense and in terms of their interactions with Lebanese people. All of the participants shared a desire to belong—to have a job, to be accepted by family, to have intimate partners, to receive non-judgmental health care services, to be able to walk down the street and take a taxi without being harassed or rejected. This positionality is important to consider when developing HIV care and prevention for these groups. Asking people who are already living at the margins to further complicate the intersecting oppressions and stigmas in their lives by taking an HIV test, or talking to their sex and drug-use partners about HIV risk, is a huge ask that could actually place them in considerable danger.

Participants described how health promotion behaviors were most available to them in the context of community, when suggested by a trusted, empathetic professional counselor or family member. These findings endorse the continuation and amplification of community-based interventions that decrease stigma by normalizing both substance use and treatment. If we are to engage vulnerable populations in HIV prevention and care, deliberate effort must be made to create inclusive communities where people feel they belong. Policies which create strict conditions under which membership can be sustained should be interrogated. Comprehensive services (i.e., support for housing, employment, and food) provided alongside substance use and HIV-related services, recognize the basic needs and dignity of clients and reject larger social dynamics that insist on seeing these communities only as drug users or refugees. Including marginalized community as community advisors and consultants—giving them a place at the table—can strengthen agency functions by including the expertise of clients.

### 4.4. Limitations

There are several limitations to this study. For one, the study did not employ any screening tool to evaluate whether respondents could be classified as experiencing substance use disorder or to clearly quantify their level of substance consumption. As a result, the study grouped together the experiences of participants with very different consumption patterns, including participants who only made recreational use of substances on an occasional basis as well as others who reported heavy and frequent consumption. Quantitative research on a larger scale, using standardized screening tools, is needed to estimate the prevalence of substance use disorder in Lebanon, both among citizens and permanent residents as well as refugees. Secondly, most (89%) of the participants in this study were men. Studies that focus exclusively on women, including transgender women, are needed to articulate and understand their unique strengths and challenges.

## 5. Conclusions

People who are surviving acute socio-economic crises, civil war, and forced migration are under extreme stress. This stress can increase substance use and decrease access to substance use treatment and HIV prevention and care services. These disruptions can also complicate sexual behaviors as regular social patterns are disrupted, privacy is curbed, and poverty provokes transactional sex. People may be increasingly isolated as discretionary pocket money, social opportunities, and public transportation dissolve. Stigma and distrust thrive as people are forced to share limited resources. In all of these ways, socio-economic crises, civil war, and forced migration can be understood as structural determinants of HIV risk. We know the interventions that work: needs assessment, education (including frank discussion about substance use and sexual behaviors), and community-building. These findings suggest that HIV education and care services would be well received when they are placed within a larger constellation of non-judgmental basic-needs services that communicate to clients “you belong here”.

## Data Availability

The data presented in this study are available in this article and Appendix A.

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
