# Peer review of "Stress, Marginalization, and Disruption: A Qualitative Rapid Situational Assessment of Substance Users and HIV Risk in Lebanon"

_ijerph, 2022, doi:10.3390/ijerph19159242_

Round 1

Reviewer 1 Report

A qualitative study that focusses on substance use and HIV risk in Lebanese communities has the potential to make a valuable contribution to understanding and addressing the factors that impact on these patterns in these communities. This is indeed a very important study, specifically at this point in the socio-political history of this region. Additionally, the restrictions placed on these already under-resourced communities during the COVID-19 pandemic have amplified the disruption and related socio-economic effects.

A study design that includes a Rapid Situational Assessment seems appropriate for a study of this nature and this is well motivated for in the manuscript. More details on the dynamics of this method is needed in the methodology section though. The COREQ criteria are also introduced, but these are not fully explained or motivated for and it is not clear how these criteria are applied in this study.

The design section makes reference to knowledge and attitudes about hepatitis which does not seem to have been explored in this study based on the findings reported and the dataset uploaded as a supplementary document. The overall aim and the specific objectives of the study could be more clearly defined. More clarity with regard to the objectives will also serve to strengthen the rationale for this study.

Based on the results reported and the dataset uploaded, it is clear that drug use patterns were thoroughly explored, however, the other two stated objectives do not seem to have been thoroughly explored and not much is reported with regard to the participants’ HIV knowledge and their perceptions and access to treatment.

It is also not clear from the results reported and the dataset uploaded whether pseudonyms were used to protect the identity of the participants. This should be stated clearly in compliance with the POPI act especially since the organisations are fully described and can be easily identified.

Based on the sampling through the two sites, perhaps the population for the study should be defined differently. This should also be reflected in the title for the study, especially considering that the data was collected qualitatively on a relatively small sample. This speaks to the representativeness of the findings.

Author Response

RESPONSES TO REVIEWER #1 COMMENTS

  1. “More details on the dynamics of this [RSA] method is needed in the methodology section.”

#1 RESPONSE: Language added to lines 113-116 to clarify the RSA method.

A citation was added (UDCCP, 1999) to offer reference to a 55-page document that describes the method in detail. The full citation was also added to the Reference List (#25). A sentence was also added to the existing description: An RSA can be used to answer a wide variety of research questions, from assessment to evaluations, generating information “from multiple sources as quickly as possible” (UDCCP, 1999, p. 7).

  1. “The COREQ criteria are also introduced, but these are not fully explained or motivated for and it is not clear how these criteria are applied in this study.”

#2 RESPONSE: Language added to lines 95-98 to clarify and define the COREQ criteria.

The COREQ (Consolidated Criteria for Reporting Qualitative Studies) checklist, “developed to promote explicit and comprehensive reporting of qualitative studies” criteria are used here to describe the project’s methodologies (Tong et al., 2007, p. 356).

  1. “The design section makes reference to knowledge and attitudes about hepatitis which does not seem to have been explored in this study based on the findings reported and the dataset uploaded as a supplementary document.”

#3 RESPONSE: Language about hepatitis deleted from lines 118, 200, 212

As the reviewer describes, the paper does make reference to hepatitis at several points in the methods section, but findings are not reported about this topic. This is an excellent point. The interview instrument did ask about hepatitis, however, data about this subject is not reported here in order to align the focus of this article with the priorities of this special issue. To avoid confusion, reference to these aspects of the project design that are not included or explored in this paper have been removed.

  1. “The overall aim and the specific objectives of the study could be more clearly defined. More clarity with regard to the objectives will also serve to strengthen the rationale for this study.”

#6 RESPONSE:

Section 1.3 was modified to more clearly articulate the study goals.

  1. Based on the results reported and the dataset uploaded, it is clear that drug use patterns were thoroughly explored, however, the other two stated objectives do not seem to have been thoroughly explored and not much is reported with regard to the participants’ HIV knowledge and their perceptions and access to treatment.

#5 RESPONSE:

Thank you for this input. The sample demographics, including information about substance use, access to treatment and HIV knowledge, are included on pages 5 and 6. We appreciate that these descriptive statistics provide limited detail, but this is what was available to us. Inferential statistics cannot be derived from this study data given the size and diversity of the sample and the semi-structured design of the instrument.

  1. It is also not clear from the results reported and the dataset uploaded whether pseudonyms were used to protect the identity of the participants. This should be stated clearly in compliance with the POPI act especially since the organisations are fully described and can be easily identified.

#6 RESPONSE: Language was added at lines 176-177:

Thank you for raising this important point. The following language has been added (Line 176) The names used in presenting the data are all pseudonyms assigned by the authors.

  1. Based on the sampling through the two sites, perhaps the population for the study should be defined differently. This should also be reflected in the title for the study, especially considering that the data was collected qualitatively on a relatively small sample. This speaks to the representativeness of the findings.

#7 RESPONSE:

Thank you for this feedback. We appreciate that the sampling process in this study was complex and have presented a detailed explanation of the recruitment process (Section 2.2). We are not exactly clear what this comment is asking us to do. We do feel the title is appropriate as written. If the reviewer would like to further clarify the comment, we will try to address it.

In a qualitative analysis, no claims to “representativeness” or generalization are made.

Additional minor edits that were not mentioned by reviews but were noticed during our review.

  • Line 5 = Middle initial added to second author
  • Line 11 = Stray underline (delete)
  • Line 22 = Word added “risk”
  • Line 24 = Keywords – “Substance” was misspelled
  • Line 274 = Word “transmitted” was misspelled
  • Line 580 = First author’s last name is misspelled

Reviewer 2 Report

The manuscript "Stress, Marginalization, and Disruption:  A Qualitative Rapid Situational Assessment of Substance Use and HIV Risk in Lebanon" is one of the best-written manuscripts that I have reviewed in a long time.   The background was carefully constructed and the case for the study was well supported. The approaches of how to reach out to this highly vulnerable group were very carefully described, as were the questions, and the analytical steps. The results were presented so that every reader in highly engaged to read their accounts, and this was followed by a detailed discussion of what needed to happen next. The attached supplementary material is fantastic and allows any reader to retrace their steps, and they have excellent references supporting this work. I have no other suggestions for this group of researchers to improve upon this article.

Author Response

Thank you very much for your generous and enthusiastic feedback. This is greatly appreciated!

This manuscript is a resubmission of an earlier submission. The following is a list of the peer review reports and author responses from that submission.